# Patient Safety in Palliative Care at the End of Life from the Perspective of Complex Thinking

**DOI:** 10.3390/healthcare11142030

**Published:** 2023-07-15

**Authors:** Nair Caroline Cavalcanti de Mendonça Bittencourt, Sabrina da Costa Machado Duarte, Sonia Silva Marcon, Marléa Crescêncio Chagas, Audrei Castro Telles, Eunice Maria Casimiro dos Santos Sá, Marcelle Miranda da Silva

**Affiliations:** 1Departament of Nursing Methodology, Escola de Enfermagem Anna Nery, Universidade Federal do Rio de Janeiro, Rio de Janeiro 20211-110, RJ, Brazil; ncarolinne@yahoo.com.br (N.C.C.d.M.B.); sabrina.cmduarte@gmail.com (S.d.C.M.D.); marleachagas@eean.ufrj.br (M.C.C.); audrei.costa@inca.gov.br (A.C.T.); 2Nursing Departament, Universidade Estadual de Maringá, Maringá 87029-900, PR, Brazil; soniasilva.marcon@gmail.com; 3Department of Medical-Surgical/Adult and Elderly, Escola Superior de Enfermagem de Lisboa, 1600-190 Lisboa, Portugal; esa@esel.pt; 4Nursing Research, Innovation and Development Centre of Lisbon (CIDNUR), 1600-190 Lisboa, Portugal

**Keywords:** patient safety, palliative care, post-modernism, nursing

## Abstract

Actions for patient safety at the end of life must be aligned with the principles of palliative care, such as promoting comfort and quality of life. Faced with this complex process, health professionals need to seek the central relationships of the concepts of safety and palliative care to the end of life, in line with the wishes and expectations of the person and family members/caregivers, as well as with available resources and the capacity of services but, above all, reinforcing the importance of a non-reductionist care approach, which encompasses the various aspects inherent to humans. Hence, we present a new vision of patient safety in palliative care at the end of life based on the complex thinking of Edgar Morin, scientific evidence, and health policies in the global context. We discuss the deficiencies and disjunctions in thought and practice of palliative care at the end of life and patient safety, as well as the challenges for the conjunction of these complex themes, to finally present potential ways to apply complex thinking in the safe care of the patient at the end of life. The problematization of different aspects for the interposition of knowledge about patient safety in palliative care at the end of life portrays the existence of intersubjective connections and the multidimensionality that permeate the guidelines, actions and relationships that sustain the disciplines.

## 1. Introduction

Worldwide, the epidemiology of the causes of death from disease is changing; non-communicable diseases and injuries are among the main causes of illness and death in the population, and living with advanced age and long-term chronic diseases leads to more complex demands for health care, such as palliative care. Palliative care is defined as “active holistic care of individuals across all ages with serious health-related suffering due to severe illness and especially of those near the end of life. It aims to improve the quality of life of patients, their families and their caregivers” [1].

When a disease threatens life because therapy with the intention of healing is no longer possible, there is a need to change the focus, offering the person a treatment aimed beyond the mitigation of physical suffering, focusing on the proper management of signs and symptoms. Therefore, it is essential to meet human needs with a multidisciplinary and comprehensive approach, through the commitment of health professionals in teamwork to humanize care, which includes accompanying family members in mourning [2].

In this sense, Cicely Saunders worked to develop this new modality of care. She was considered the pioneer of the modern hospice movement, making palliative care official as a distinct practice in the health area in the 1960s in the United Kingdom. Born in England in 1918, she graduated in nursing, social work, and later in medicine, having her reflections based on these different experiences and professional skills. At the time, most institutions for end-of-life patients focused on religious, philanthropic, and moral aspects, with little involvement in health care. By qualifying as a physician, Cicely Saunders sought to integrate clinical practice with research and teaching in favor of scientific rigor, essential for the beginning of the transformation of caring for patients at the end of life [3].

Her reflections regarding the multidimensional aspects of pain in her publications cover suffering with attention to listening and understanding the experience of suffering with an integral approach. The careful analysis of the records after attentive listening made it possible to identify the physical, emotional, social and spiritual dimensions, highlighting issues related to the need for security and values that transcend everyday life, reinforcing the importance of a non-reductionist care approach, which encompasses the various inherent aspects to humans [3].

Since the beginning of the 21st century, the palliative care movement has been growing worldwide, but access to such care is still deficient, and many times restricted to the end of life. Palliative care is a global public health emergency, and the inequality of access, quality, and safety of palliative care among countries has mobilized efforts from government entities, such as the World Health Organization (WHO), to encourage the development of palliative care in response to the complexities of health systems and care needs of people and their human rights. Palliative care is more developed and integrated into the health systems of high-income countries, while of the approximately 40 million people who need palliative care per year, 78% live in low- and middle-income countries [4].

According to the WHO [4], palliative care is more effective when offered from the onset of the disease, as its timely provision not only improves the quality of life of patients but also reduces the use of emergency services and hospital admissions and, consequently, health costs, based on planning that facilitates person-centered care. However, it should be noted that the complexity of care demands tends to increase with the progression of a disease and, although palliative care is fundamental in all stages of the disease in response to suffering, it gains even more emphasis at the end of life, when not more associated with systemic treatments to control the disease, based on acceptance of the natural evolution of the disease.

However, if palliative approaches take place from the onset of the disease, it becomes easier to recognize the different complexities of needs and ensure greater security in referrals to specialized palliative care in more complex cases, whether in the physical and/or psychosocial and emotional aspects. Specialized palliative care systematically seeks to assess symptom control scores, quality of life, and capacity function, committing resources to provide a dignified and comfortable death, in line with the wishes and expectations of the person and family members/caregivers [5].

The characteristics of the person at the end of life increase the exposure to the risk of adverse events related to multiple factors, such as loss of muscle strength, restriction to bed, dependence on care, cachexia, lowering of the level of consciousness, delirium, psychomotor agitation, skin fragility and presence wounds, use of sedatives and opioids, use of invasive devices, and vulnerability to social isolation and family abandonment, among other issues as the disease progresses [6].

Notoriously, the multidimensional fragility of the end-of-life person should be valued, since the occurrence of adverse events resulting from health care can cause several harms, including to family members/caregivers, and health professionals. Given this, it is essential to discuss patient safety, which is defined as “a framework of organized activities that creates cultures, processes, procedures, behaviors, technologies and environments in health care that consistently and sustainably lower risks, reduce the occurrence of avoidable harm, make error less likely and reduce its impact when it does occur” [7].

Concern for patient safety while receiving medical care has been known since Hippocrates (460 to 370 BC) stated “primum non nocere”, which means “first, do no harm”, demonstrating the perception that care could cause some type of damage. And, in this same sense, Florence Nightingale, in the 19th century, formulated the reflection: “perhaps it may seem a strange principle to enunciate as the first duty of a hospital not to cause harm to the patient”, structuring a care model in the Crimean War, from the attention in the separation of soldiers according to the type of illness, and in the implementation of improvements in housing and care for their hygiene and comfort [8].

However, safety in the health area only received focus in 1999, after the publication of the study “To err is Human: building a safer health system”, which presented the problem of adverse events and patient safety in general, consequently leading to greater attention from the media and health professionals around the world [9].

With the need to increase global attention to patient safety issues, it became necessary to strengthen the discussion on the subject and establish norms and standards. To this end, the WHO created, in 2004, the World Alliance for Patient Safety, involving regulatory agencies, governments, and patients, with the objectives of organizing concepts and defining patient safety, in addition to proposing measures to reduce risks and events adverse [10].

From this Alliance, the WHO Safety and Risk Management Unit was formed, which culminated in the creation of the Global Patient Safety Challenges, a program aimed at identifying the most significant risks to health, and then developing tools and strategies for damage prevention. The WHO provides leadership and guidance in collaboration with member countries, stakeholders, and experts to develop and implement interventions and tools to mitigate risks, improve safety and facilitate change [11].

Patient safety should be considered a strategic priority for modern health care, as it is already recognized that every point in the care process may contain an inherent risk; however, the nature and scale of risks vary according to the context of health care delivery, its availability, infrastructure, and resources within and between countries. Patient safety encompasses organized activities with the creation of cultures, processes, procedures, behaviors, technologies, and environments in health care, consistent and sustainable, to reduce risks and avoidable damages, making errors less likely and have less impact [7].

However, even after many years, more maturity is still needed on this subject, since the occurrence of adverse events due to unsafe care is probably one of the ten main causes of death and disability worldwide [12]. We start from the hypothesis, which has guided investigations in our research group, that the context of palliative care is more vulnerable to risk; whether at home or more intensely in the hospital, the person with a serious illness has varied and complex reasons that increase this exposure, mainly related to the difficulty of communicating bad news; sharing decision-making, whether in the health team, and with patients and family members/caregivers; and, above all, mismanagement of multidimensional symptoms [13].

It should be added that dealing with patients in palliative care at the end of life leads to a lack of time, a challenge to be managed so that the sick person can find psycho-emotional balance, rescue relationships, and time must be respected for the quality of life to death, rather than being shortened due to the consequences of preventable harm. It is common for these people to need time to be with family and friends, and resolve postponed conflicts and any other issues responsible for anguish, sadness and depression, which requires adequate control of physical symptoms [14].

In this sense, we must also draw attention to the time required to recover from possible damage resulting from care failures, making issues involving patient safety in palliative care at the end of life of fundamental importance for raising discussions that can stimulate the production of scientific evidence to assist in the direction of conduct and mitigation of impact on quality of life.

Therefore, patient safety needs to be considered a strategic priority for health institutions, recognizing the potential risk for the occurrence of adverse events related to health care and, for it to be effective, it must be aligned with the particularities of the context. Currently, approximately 134 million adverse events related to unsafe care occur in hospitals in low- and middle-income countries, and most end-of-life patients die in hospitals [7].

It is noteworthy that it is in these countries where palliative care, for the most part, is less developed and has less availability of resources, including the education of professionals and the literacy of the population on this subject. The development of palliative care programs is correlated with the level of the Human Development Index, characterized by slow growth over the years and international inequality [15]. The lack of knowledge and lack of necessary resources have an impact on meeting the guiding principles and, ergo, on recognizing and responding to the needs of the patient and family members, generating a negative reflection on the quality and safety of care.

It becomes necessary to understand patient safety in palliative care at the end of life as a complex phenomenon, stimulating interdisciplinarity interaction, and articulation of different knowledge with the interposition of principles. Therefore, there will be the reach of a constant problematizing thought, which allows to intervene, lead, and redirect in a cyclical, changeable movement, which establishes a dialogue with the real [16].

Hence, this perspective presents a new vision of patient safety in palliative care at the end of life based on the complex thinking of Edgar Morin [16,17,18], scientific evidence, and health policies in the global context. We discuss the deficiencies and disjunctions in thought and practice of palliative care at the end of life and patient safety, as well as the challenges for the conjunction of these complex themes, to finally present potential ways to apply complex thinking in the safe care of the patient at the end of life.

## 2. The Current Situation of Palliative Care at the End of Life and Patient Safety: Evidence of Deficiencies and Disjunctions in Thought and Practice

Complex thinking emerges in an antagonistic way towards simplifying thinking, refusing mutilating and one-dimensional consequences of thinking, with the ambition of articulating dismembered disciplinary fields, even knowing that complete knowledge is impossible, in which one of the axioms is the impossibility of omniscience [17]. In this sense, the forwarding of this discussion aspires to a dialogue based on the identification of links between palliative care at the end of life and patient safety, intending to motivate multidimensional reflective thinking.

Palliative care and the relief of suffering are among some of the most neglected dimensions in global health, causing people with serious illnesses to experience an extreme burden of suffering [4] and, in parallel, the path of developing patient safety over time, it has been marked by the omission of warnings about the size of the problem, also characterized by the slow pace of integration into the daily work of institutions and the culture of professionals [19].

It is estimated that, worldwide, more than 56.8 million people need palliative care every year; among these, 25.7 million are close to the end of life [15]. However, depending on the disease and other factors related to the person, it can be difficult to predict this stage, leading some people to receive palliative care only in the last weeks, days, or hours of life. WHO data indicate that the global need for palliative care will continue to grow as a result of populations aging and the growing burden of non-communicable diseases and some communicable diseases. Based on this scenario, we argue that patient safety at the end of life requires access to quality palliative care, wherever it is. Not having access to palliative care already defines unsafe care in situations of serious illness [4].

Evidence of global health points to disrespect for the guarantee of universal access to safe, dignified end-of-life care. A portrait of the behavior of our societies as non-trivial machines, who know without ceasing political, economic and social crises, makes it necessary to invent strategies to get out of the crisis, develop new solutions, and abandon the old ones [17].

It is noted that dignity and security at the end of life have not yet become health priorities, a situation related to health outcome measures based on the biomedical model, which still guides the main drivers of policies and investments and focuses on extending life and productivity [20], the result of a hegemonic model based on the Cartesian paradigm, characterized by a linear, disciplinary and hierarchical organization of knowledge that closes in on itself, not admitting contact with other ideas and favoring overspecialization to the detriment of a multi-referential and transdisciplinary approach [16].

The domain of the biomedical model, with a focus on healing, contributes to the seeing death by health professionals as a failure of care and not as an opportunity to direct towards a different type of care, with a person-centered approach, and grounded in comfort and dignity [13].

## 3. The Challenges for the Conjunction of End-of-Life Palliative Care and Patient Safety: Contributions from Complex Thinking

Against the grain of the modern world, palliative care demands the connection of knowledge and, based on the paradigm of complexity, brings the appreciation of interrelationships, understanding that health is a multidimensional process that needs a complex look to be understood, contemplating the union of different areas of professional activity, knowledge, dimensions, and their interdependence [21].

In this conjecture, an effort is needed to increase the supply of palliative care globally, on the part of public policymakers, decision-makers, and health care professionals who provide care, including nurses, who are a vital part of the multidisciplinary team. Therefore, health professionals must be qualified both to offer comprehensive and complex care and to provide knowledge to patients and their relatives, participating in the formulation of individualized care plans and acting as patient advocates [22]. In addition, there is an incentive to recognize patient safety as a strategic priority in health institutions and actions based on safe practices. Admittedly, there is complex work in the promotion of a transdisciplinary approach, in which the anthropological refers to the biological, which refers to the physical, which refers to the anthropological, in a circular communication between the sciences [18].

Initially, it is necessary to obtain information to support safe work processes that receive the fragility of patients under palliative care, considering that they are more vulnerable to incidents related to patient safety, and still little is known about the extent of damage caused or the origins of unsafe care in this population [23]. A fact that presents itself as complex, in the sense that to reach knowledge, phenomena must be aligned, rejecting disorder and removing uncertainty, occurs through the selection of elements such as clarifying, distinguishing, and hierarchizing [17].

The international standard ISO 31000 establishes some elements that contribute to patient safety in health services, such as the need for integrated, structured, comprehensive, personalized, inclusive, dynamic care, with the best information available, and adequate human and cultural factors in pursuit of continuous improvement in health care [24]. In this sense, the Global Action Plan for Patient Safety [7] highlights the importance of patient involvement in their care through Strategic Objective 4. On the other hand, it is known that multiple contributing factors result from circumstances, actions, or omissions that have undesirable results during the provision of care [25].

These results are titled in the patient safety literature as: a near miss, which is an incident did not reach the patient; a no harm incident, which is one in an event reached a patient but no discernable harm resulted; and an adverse event, which is an incident that results in harm to a patient. This harm implies impairment of the structure or function of the body which a deleterious effect, including disease, injury, suffering, disability, and death, and may be physical, social, or psychological [26].

In order to prevent these avoidable and undesirable results, systematic risk management is indicated, which includes, among its stages, the performance of analyses to define the degree of risk, which includes quantifying the probability of occurrence and the associated severity; however, often the damage identified coincides with possible implications for advanced disease in the person at the end of life, which makes this analysis complex [26].

Complexity makes us prudent, attentive, and helps us recognize uncertainty and avoid contradictions; this implies overcoming, changing the absolute character of “either this/or that” and the quantity/or quality, based on a dialectic between analytical–reductionist thinking and global thinking, thus removing the damage resulting from the effects of fragmented and unidimensional thinking [17].

Therefore, skills must be developed to identify and raise awareness of the dangers associated with health care in the context, a safety precept that involves, above all, the human factor, related to the ability, competence and attitude to know how to recognize threats. This knowledge is complex not only because of behaviors shaped by different ways of building technical knowledge but also associated with sociocultural issues, which are decisive for the reform of thought, and for the change in the ways of knowing, thinking and acting [18].

In addition, such elements that increase patient safety in palliative care at the end of life must be applied in line with the principles to achieve comfort promotion, the main goal of this approach, making it essential to pay attention beyond the relationship of the different disciplines addressed to the relationship ability between the different individuals. Complex processes are those in which the concepts should never be defined from their limits, but rather search for the definition of a center, in the understanding of the existence of mixed intermediaries [17].

This complex way of thinking also refers to which is woven of heterogeneous and associative constituents, allowing distinguishing without separating, and association without identifying or reducing [17], which produces a foundation for the integral approach to the sick person, fundamental to palliative care at the end of life, which requires a multidisciplinary team working in an interdisciplinary way, the combination of knowledge and actions being fundamental, a condition that coincides with precepts of patient safety.

Discomfort can trigger another uncomfortable sensation, which may not be related to physical pain, which requires specific knowledge to understand the difference between pain relief and total comfort [3]. The health team must consider and manage the physical, social, psychological, and spiritual dimensions, essential for patient safety in palliative care at the end of life, to provide support to reduce suffering. In this sense, the complexity approach collaborates with the necessary reflection to contemplate treatment possibilities, as a way to qualify life, learn to live and die with dignity, understand the multidimensionality of human existence, respect and approach grief, and recommend permanent and interdisciplinary professional education [27].

## 4. Potential Ways to Apply Complex Thinking in the Safe Care of the Patient at the End of Life

Complementary to teamwork, converges between the guidelines of the disciplines on the screen and patients and family members being involved and included as full partners in care, provides the promotion of autonomy and instrumentalization for the shared decision-making. People’s reports represent a different perspective on safety during health care based on their experiences, which are generally not taken, but can serve as a basis for improving the quality of health care and shared decision-making [25]. A new vision, aggregating information, in line with complex thinking that aspires to multidimensional knowledge [17].

For this, the work at the front of the professional in health education must be strong, seeking to reach society for greater health literacy on palliative care, present from the moment of diagnosis, during the evolution of the disease, until the care at the end of life [7], as well as on patient safety precepts. At that moment, the complexity in the construction of knowledge through dialogic is considered, making it possible to contextualize the object, favoring articulations, without eliminating differences, respecting the particularities of the parts [17].

New information reported by patients and family members is valuable, even when problems in the safety of care overlap with what is perceived by health professionals, as they can broaden the understanding of the magnitude and factors contributing to the occurrence of these problems [25]. However, it must be considered that any knowledge operates by selecting significant data and rejecting non-significant data, attending to commands based on “supra logical” principles of organization of thought or paradigms, hidden principles that govern our view of things and the world that we still do not know [17]. Thus, the individual’s previous experiences can influence the perceptive process and, consequently, the decision-making process, which involves the professional in addition to the patient/family.

There is also complexity in the weave of events, actions, interactions, retroactions, determinations, and chance, which constitute our phenomenological world [17]; a reflection of the contribution of knowledge by patients, family members, and caregivers from experiences that are not experienced and cannot be reproduced by health professionals, considering the particularities and cultural aspects of each individual.

Furthermore, in palliative care at the end of life, for risk management to be effective, it is fundamental to receive the family’s demands. Thus, in addition to including the response to the family’s concerns at the end of life, support after the patient’s death should be included, which also involves an invitation to become involved in post-death care [6]. The conception of these different domains strengthens complex thinking, which brings with it its subjectivity, its mutability [17] and, in this sense, the freedom to rethink programs, strategies, and new solutions to problems related to patient safety in the context of palliative care at the end of life.

It should be considered that some patients, especially older and less educated ones, may prefer not to know and not to participate in the decision-making process, delegating decisions to health professionals or the family, in contradiction to the discourse that decisions should include patients and/or family members [28].

For better results, the education of health professionals, particularly nurses, permeated by new concepts, terms and references, needs to encourage a critical and reflective profile, making the reductionist view of the world insufficient for current times. The use of complexity theory favors daily reflection, questioning, and social transformation, based on thought capable of considering all influences and emphasizing uncertainty as a guiding principle of humanity, as it suggests that one seeks to understand the contradiction and the unpredictable, based on the coexistence of these two aspects [29].

## 5. Conclusions

We believe that reflecting on patient safety in palliative care at the end of life in light of the complexity, favors a differentiated perspective for expanding knowledge on the subject, contributing to the awareness of health professionals regarding the recognition of intersubjective connections, and the appreciation of multidimensionality that permeates the guidelines, actions, and relationships.

This reflection allowed the revealing of relational and interactive processes of patient safety in palliative care at the end of life. It was understood the transdisciplinary participation of the team and that, in addition to the focus on the sick person, it is essential to recognize the family as a care unit, because, despite the suffering resulting from its emotional involvement, they actively participate throughout the process and can contribute with knowledge from their experiences.

The recognition of the uniqueness involved in patient safety in palliative care at the end of life supports the elaboration of tailored care plans, allowing better care management and preventing unintended actions that generate suffering.

We recognize the limitation of reflection influenced by the world view of the authors; however, the importance of new research to focus on the theme addressed from the recognition of the complexity involved and the significant impact on people’s lives and on health systems can be stated.

## Data Availability

No new data were created or analyzed in this study. Data sharing is not applicable to this article.

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
