# Peer review of "Patient Safety in Palliative Care at the End of Life from the Perspective of Complex Thinking"

_healthcare, 2023, doi:10.3390/healthcare11142030_

Round 1

Reviewer 1 Report

I think this study is a good paper related to Quality and Patient Safety in Palliative Care. However, please explain the method part of the study by inserting a table or figure for the reader.

Author Response

Manuscript ID: healthcare-2464508

Patient safety in palliative care at the end-of-life from the perspective of complex thinking

Reviewer 1:

Comments:

I think this study is a good paper related to Quality and Patient Safety in Palliative Care. However, please explain the method part of the study by inserting a table or figure for the reader.

Response:

Thank you for taking the time to read our manuscript and thus contributing to the scientific publishing process. In line with what was also commented on by reviewer 2, we have included information about the non-systematized review. Even so, we kept it in the form of a text, which we consider not to harm the understanding and didactics of the manuscript.

Reviewer 2 Report

Thank you for the opportunity to read the interesting article. It suggests specifying the description of the research methodology, indicating the databases from which the analyzed scientific articles were obtained.

The reviewed manuscript "Patient safety in palliative care at the end-of-life from the perspective of complex thinking" is an interesting study of the subject matter. The initial part of the work contains an introduction to the history of palliative care initiated by Cicely Sunders and issues related to patient safety.  

The manuscript  was written on the basis of an unstructured bibliographical review. I would suggest including in the text information on the criteria for including publications in the analysis

- what guided the selection of literature (key words?),

- what databases were searched?

- what period of time did the source publications come from?

- how the articles were selected for analysis (reaching a consensus among the authors? One author decided? Each of the authors gave their suggestions?)

Author Response

Manuscript ID: healthcare-2464508

Patient safety in palliative care at the end-of-life from the perspective of complex thinking

Reviewer 2:

Comments:

The manuscript was written on the basis of an unstructured bibliographical review. I would suggest including in the text information on the criteria for including publications in the analysis

- what guided the selection of literature (key words?),

- what databases were searched?

- what period of time did the source publications come from?

- how the articles were selected for analysis (reaching a consensus among the authors? One author decided? Each of the authors gave their suggestions?)

Response:

We thank you for reading our manuscript and comments with questions that will help clarify the procedure for selecting the conceptual bases on the topic. All responses were included in the second paragraph of the method, highlighted.

Reviewer 3 Report

The objective of this paper is to present a theoretical-reflective essay based on data from official websites and other sources, on patient safety in palliative care at the end of life.

Specific analysis and tools do not support the presented method.

The considerations specified in the paper must be supported by a detailed analysis of the elements identified in the mentioned studies.

An analysis of the collected data is required to support the statements in this work.

Some text sequences can be found in papers that are not referenced, such as:

- GLOBAL PATIENT SAFETY ACTION PLAN 2021–2030, Towards eliminating avoidable harm in health care

https://doi.org/10.3390/ijerph20126121

-  doi: 10.1177/0269216318817692.

The authors should express the analysis in a more precise and usable way for the reader.

Author Response

Manuscript ID: healthcare-2464508

Patient safety in palliative care at the end-of-life from the perspective of complex thinking

Reviewer 3:

Thank you for taking the time to read our manuscript and thus contribute to the scientific publication process, which is so complex. We are especially grateful for the critical and careful analysis, which we believe that by seeking to meet the points of this review, we had the opportunity to qualify our manuscript.

Comments:

Point 1: Specific analysis and tools do not support the presented method.

Response 1: We sought to enrich the method with information about the non-systematized review, and we believe that all conceptual and theoretical bases consulted supported the discussion on the proposed topic.

Point 2: The considerations specified in the paper must be supported by a detailed analysis of the elements identified in the mentioned studies. An analysis of the collected data is required to support the statements in this work. The authors should express the analysis in a more precise and usable way for the reader.

Response 2: With these comments, we understand the need to reorganize the discussion, replacing this generalized title with more specific topics, in order to direct the reader to the authors' intentions for approaching the evolution of the theme and future needs.

Point 3: Some text sequences can be found in papers that are not referenced, such as:

- GLOBAL PATIENT SAFETY ACTION PLAN 2021–2030, Towards eliminating avoidable harm in health care

- https://doi.org/10.3390/ijerph20126121

-  doi: 10.1177/0269216318817692.

Response 3: These articles were not selected in our non-systematized review. We checked its content and identified that we did not structure our discussion based on the results of these studies. On the other hand, convergences can be identified in their backgrounds, but this also applies to the other selected studies. And the Global Patient Safety Action Plan 2021-2023 is number 7 of our references.

Reviewer 4 Report

- What is complex thinking?

- Well known information is repeated.

- There is a disorganization in terms of subject integrity.

- Who is team, what is teamwork?

Author Response

Manuscript ID: healthcare-2464508

Patient safety in palliative care at the end-of-life from the perspective of complex thinking

Thank you for taking the time to read our manuscript and thus contribute to the scientific publication process, which is so complex. We are especially grateful for the critical and careful analysis, which we believe that by seeking to meet the points of this review, we had the opportunity to qualify our manuscript.

Reviewer 4:

Comment 1: What is complex thinking?

Response 1: The definition is in the first paragraph on page 4.

Comment 2: Well-known information is repeated.

Response 2: As these are issues that still have important gaps in practice, education, and policy, which can result from several causes, among them the difficulty of knowledge translation, we chose to maintain well-defined concepts in the literature in the structure of the text, also considering the transversality of the public-target that we intend to achieve with this manuscript, in addition to its reflective potential to contribute to the health literacy of the population on such sensitive topics.

Comment 3: Who is team, what is teamwork?

Response 3: In this case, all concepts worked on in the manuscript highlight the importance of teamwork to reach your goals, such as palliative care, patient safety, and complex thinking. We included more details about teamwork in the penultimate paragraph on page 1, but we chose not to detail which professionals should compose this team, because despite there being a recommendation for its standard composition, also well presented in the literature, each situation requires a certain approach through individualized care.

Round 2

Reviewer 3 Report

I appreciate the authors' response to my remarks and suggestions, but, nevertheless, I believe that the scientific work, according to the theme of the journal, should be supported by a formalization through tables, values, and research analyses. These were not introduced in the revised version.

For example, the abstract mentions “Methods:  theoretical-reflective essay based on data from official websites on the 17 subject, scientific articles and books, accessed between January 2022 and March 2023”.  The data could be presented formally in tables.

Author Response

Manuscript ID: healthcare-2464508

Patient safety in palliative care at the end-of-life from the perspective of complex thinking

Reviewer 3:

Comments:

Point 1: I appreciate the authors' response to my remarks and suggestions, but, nevertheless, I believe that the scientific work, according to the theme of the journal, should be supported by a formalization through tables, values, and research analyses. These were not introduced in the revised version.

For example, the abstract mentions “Methods:  theoretical-reflective essay based on data from official websites on the 17 subject, scientific articles and books, accessed between January 2022 and March 2023”.  The data could be presented formally in tables.

Response 1: Thank you again for taking the time to read our manuscript and thus contribute to the scientific publication process. In view of the better framing of the typology of this manuscript, as a perspective, and the guidelines for authors, we realized the need to exclude the materials and methods section, and we incorporated in our objective the theoretical/conceptual bases that supported its construction. We have also modified the abstract.

Reviewer 4 Report

Thank you.

Author Response

Manuscript ID: healthcare-2464508

Patient safety in palliative care at the end-of-life from the perspective of complex thinking

Reviewer 4:

We thank you once again for your availability to read and analyze our manuscript. We made changes to our abstract and objective, and we excluded the materials and methods section due to the understanding to better fit the typology of the manuscript - perspective. As there was no specific comment in your revision, we hope to have taken care of it. Best regards, the authors.